# Toward an integrated model of dietary behavior: Social determinants and adherence to the mediterranean diet

**Francesca Romana Lenzi** [1]*, **Ciro Clemente De Falco**[2], **Michela Cavagnuolo**[1], **Vincenzo Esposito**[2], **Ferdinando Iazzetta**[3]

**1** Department of Movement, Human and Health Sciences, University of Rome "Foro Italico", Rome, Italy, **2** Departement of Social Sciences, University of Naples "Federico II, Naples, Italy, **3** Department of History, Anthropology, Religions, Art, Performing Arts, University of Rome "La Sapienza", Rome, Italy

* francescaromana.lenzi@uniroma4.it

## Abstract

### Background

Adherence to the Mediterranean diet is widely recognized as a protective factor for health, yet its determinants remain complex and multidimensional. This study investigates the social, economic, cultural, familial, and perceptual factors associated with adherence to the Mediterranean diet within the Italian population.

### Methods

Using a stepwise approach based on four linear multiple regression models, we analyzed nationally representative survey data to assess how different sets of variables influence adherence to the Mediterranean diet. The analysis was informed by the theoretical framework of the social determinants of health, the nutrition transition theory, and Bourdieu's concept of cultural capital.

### Results

Initial models identified younger age and higher educational attainment as significant predictors of greater adherence. With the inclusion of socioeconomic variables, household income emerged as a more robust predictor, while the effect of education diminished. Subsequent models showed that family cultural capital and individual awareness of diet-health relationships significantly increased explanatory power. In the final model, perceptual and cognitive variables—especially the recognition of the role of nutrition and body weight in health—were among the strongest predictors (adjusted $R^2 = 0.25$; Durbin-Watson $= 1.98$).

**Data availability statement:** all relevant data are within the manuscript and its Supporting information files.

**Funding:** the study was funded by the Cariplo Foundation (Project "Dietary fructose in pediatric obesity-related diseases: identification of nutritional, biological, omics and social determinants"). The payment, however, will be processed by the principal investigator of the funded unit, which may have caused the discrepancy in the submission system. Therefore, we would like to clarify that the funding has been provided by the Cariplo Foundation.

**Competing interests:** The authors have declared that no competing interests exist.

## Conclusions

Adherence to the Mediterranean diet reflects not only economic or educational resources, but also deeper cultural and cognitive dimensions. The findings suggest that effective public health interventions should integrate nutritional education, cultural engagement, and targeted support for socially and geographically disadvantaged groups. Rather than a purely behavioral choice, dietary adherence should be understood as a socially structured practice embedded in broader patterns of inequality.

---

## 1. Introduction

The findings presented in this article are part of the research project *"Dietary fructose in pediatric obesity-related diseases: Identification of nutritional, biological, omics, and social determinants"*, funded by the Cariplo Foundation. Launched in March 2023, the project investigates fructose consumption in young obese patients through a multidisciplinary approach, with a sociological focus on the social and cultural determinants shaping nutritional awareness in patients and families.

This population was identified through recruitment in the clinical units based in Novara (University of Eastern Piedmont "Amedeo Avogadro") and Bari (University of Bari "Aldo Moro") and was the focus of both clinical and sociological investigations. Sociological data were collected through questionnaires administered to patients and their families by the Rome "Foro Italico" unit.

The Rome 'Foro Italico' unit also conducted an online survey on dietary behaviors, Mediterranean diet adherence, nutritional knowledge, and family contexts, to analyze how social and economic factors influence food literacy and health promotion [1].

Health is not determined solely by biological or clinical factors, but is strongly influenced by social, economic, environmental, and cultural conditions. These factors, known as the Social Determinants of Health (SDH), encompass the circumstances in which individuals are born, grow, live, work, and age—circumstances that are themselves shaped by the distribution of power, resources, and opportunities at global, national, and local levels [2]. Social determinants are primarily responsible for health inequalities: systematic, unjust, and preventable differences in health status among different population groups. The WHO (World Health Organization) has highlighted how factors such as income, education, social protection, access to basic services, living environment, and food insecurity directly affect both individual and collective health [2].

The Dahlgren and Whitehead 'Rainbow Model' [1,3] illustrates health as the result of interactions between lifestyle, social networks, and broader structural conditions (employment, housing, services, environment) (Fig 1).

This model represents health as the outcome of the interaction between individual characteristics and environmental and social factors. Consequently, an approach to health that considers only clinical indicators is insufficient. It is necessary to integrate

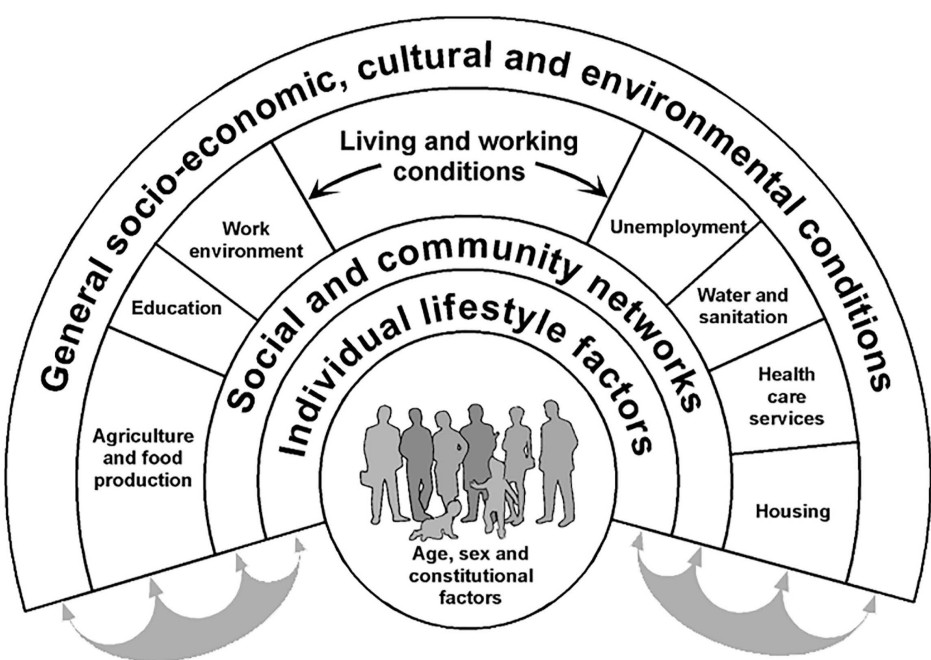

Source: adapted from Dahlgren and Whitehead, 1991

**Fig 1. Rainbow model.** Source: Dahlgren e Whitehead (1991; 2021).

the analysis of social determinants to fully understand health inequalities, prevent the worsening of vulnerabilities, and design effective and inclusive policies.

Among the dimensions most sensitive to the social determinants of health is nutrition, which both reflects and contributes to reproducing systemic inequalities. In this context, the various scientific contributions of Barry M. Popkin have played a crucial role in redefining the understanding of the nutrition transition—that is, the shift from traditional diets, characteristic of past eras (often based on fresh, seasonal, and home-cooked foods), to modern and postmodern dietary patterns dominated by ultra-processed products rich in sugars, saturated fats, and sodium. Parallel to this transition, Popkin identifies an inversely proportional trend in physical activity, which has given way to increasing sedentary behavior [4,5]. The acceleration of these trends after World War II has led to a growing pandemic of obesity and non-communicable diseases (NCDs) that has affected high-, middle-, and low-income countries alike, producing forms of malnutrition associated with obesity and its proliferation particularly among the most disadvantaged social groups [6,7]. However, this transition does not follow a uniform path. Popkin identifies distinct phases that manifest unequally both between and within countries, shaped by cultural, economic, and political variables. In many middle- and low-income economies, economic growth and rapid urbanization have produced a "double burden of malnutrition," characterized by the coexistence of chronic undernutrition (e.g., stunting and severe thinness) and obesity within the same population or even within the same household. For example, studies conducted in countries such as India, Brazil, Ghana, and China show that as income rises, diet quality often deteriorates due to the replacement of traditional foods with industrial, often imported products that are economically accessible but nutritionally poor [8]. This contemporary malnutrition increasingly takes the form of "hidden hunger," presenting not only as a quantitative deficiency but also as a caloric excess with qualitative deficits, often linked to a lack of essential micronutrients. This is a kind of nutritional paradox driven by global market logics and an economic structure that makes less healthy foods more accessible, while reserving nutritious foods for those who can afford them.

It is a phenomenon that Popkin attributes to the convergence of food industrialization dynamics, market deregulation, and neoliberal consumption models.

In high-income countries, malnutrition mainly affects vulnerable groups facing barriers to healthy food. 'Food deserts,' or areas with limited access to fresh products, create obesogenic environments that reinforce inequalities [9]. In these contexts, food also assumes symbolic and compensatory values, tied to identity, affection, or sociability, which may conflict with health-promoting and normative messages that are often perceived as elitist.

In low- and middle-income countries, uncontrolled urbanization promotes sedentary lifestyles and the erosion of collective eating practices, creating contexts where food poverty and excess coexist from early childhood [10]. In all contexts, food practices mirror social stratification not only through access but also knowledge and autonomy. Food literacy—beyond nutrient knowledge—means interpreting information and understanding the social and health implications of choices. Unevenly distributed, it reinforces inequalities [11–13]. Food is also cultural and symbolic, expressing belonging and collective memory; culture should therefore be treated as a resource for health promotion [14]. In Italy, this interweaving of food, culture, and social structure assumes particularly significant characteristics. On the one hand, the country is historically associated with the Mediterranean diet, which in 2010 was recognized by UNESCO as an Intangible Cultural Heritage of Humanity and is considered one of the most balanced and sustainable nutritional models in the world [12,15,16,17]. On the other hand, however, actual adherence to this model is progressively declining, especially among younger generations and within the most socioeconomically disadvantaged contexts [17]. In some regions of Southern Italy, for example, a growing prevalence of childhood obesity has been recorded, despite the geographical and cultural proximity to the original territories of the Mediterranean diet [16].

This paradox reveals that food culture alone is not sufficient to ensure health and well-being unless it is supported by favorable material conditions, public policies, and educational initiatives capable of valorizing and updating the local gastronomic heritage. Furthermore, the Mediterranean diet cannot be reduced to a mere set of healthy eating practices: it represents a complex cultural ecosystem, composed of traditional knowledge, intergenerational relationships, agri-food biodiversity, and environmental sustainability. From this perspective, it also functions as a social and symbolic indicator: its progressive marginalization signals not only a change in dietary behaviors but also a loss of cultural cohesion, equitable access to common goods, and continuity in collective food memories.

Although scientifically recognized for its protective effects against major chronic diseases and for its overall contribution to improving quality of life [18,19], the Mediterranean diet today risks surviving more as a symbolic imaginary than as a daily practice rooted in local territories. Increasingly, it is being reinterpreted as an elitist consumption style, accessible only to those who possess sufficient time, cultural capital, and economic resources. The ability to cook, choose, and interpret nutritional information is strongly influenced by the cultural and social capital available. Thus, food becomes a mirror of power asymmetries: those who can choose what to eat are not only those with economic means but also those with time, cognitive tools, and freedom of mobility.

Popkin [6,7] emphasizes that the nutrition transition is not a biological or economic destiny but a historical and political process, therefore subject to change. Bold public policies—such as sugar taxes, advertising restrictions, clear labeling, product reformulation, and incentives for healthy food—can slow or reverse unhealthy dietary patterns. Experiences in Chile and Mexico show that food environments can be reshaped [20,21]. Addressing malnutrition and inequalities requires both cultural and structural change: food must be recognized as a common good, requiring investment for equity and sustainability [22].

Food poverty is not simply the lack of food but the inability to access healthy and culturally appropriate diets in a stable and dignified way. It reflects economic, territorial, cultural, and social barriers with systemic public health consequences [23,24]. In high-income contexts, it persists in subtle forms, disproportionately affecting low socioeconomic groups, who face higher risks of nutritionally poor diets and adverse health outcomes [25].

Over the past two decades, research has shifted from individual choices to structural explanations of food behavior. Social and economic environments strongly shape diets, limiting the effectiveness of policies based on personal responsibility [26]. Low-income populations face economic, geographic, and cultural barriers to healthy food [27,28], producing a social gradient: as socioeconomic status decreases, diet quality systematically worsens [29].

Food inequalities are a clear expression of wider social inequalities. Disadvantaged groups are more exposed to chronic diseases linked to poor nutrition, including obesity [30,31], type 2 diabetes [32,33], cardiovascular diseases [34,35], and widespread "silent" malnutrition [36]. These outcomes result not only from individual decisions but from structural political and economic arrangements that restrict healthy choices [37].

International studies confirm the correlation between income, food insecurity, and diet quality, with higher risk behaviors among poorer groups [38,39]. Rising living costs, reduced purchasing power, weakened welfare, and widespread job insecurity have worsened these gaps, while the COVID-19 crisis further amplified food insecurity and inequalities [40,41]. However, not all vulnerable groups adopt harmful habits: adaptive mechanisms and social resources can mitigate the impact of inequalities [42].

In light of the theoretical framework outlined above, it becomes essential to explore how social determinants—particularly cultural capital, access to resources, and everyday dietary practices—translate concretely into different levels of adherence to the Mediterranean diet. The empirical investigation presented in this contribution aims to examine these dynamics by analyzing dietary behaviors within the Italian population through an integrated quantitative approach.

## 2. Materials and methods

The present study, conducted within the framework of the project "DIETARY FRUCTOSE: A Metabolic Switch in Paediatric Obesity-Related Diseases—Identification of Nutritional, Biological, Omics, and Social Determinants" (hereafter DIET-FRUCTOSE project), aims to investigate dietary habits and the level of adherence to the Mediterranean diet among the Italian population, with particular attention to social and cultural factors. To achieve this aim and build a robust empirical foundation, a web-based survey (WBS) methodology was employed. The study was conducted in accordance with the Declaration of Helsinki (2013 revision), the EU General Data Protection Regulation (Regulation (EU) 2016/679). It received ethical clearance from the Ethics Committee of the University of Rome "Foro Italico" (CAR 153/2023; approval date 28 March 2023). Data were collected between 15 September 2024 and 28 November 2024 through a Computer-Assisted Web Interview (CAWI) hosted on SurveyMonkey. Only individuals aged ≥ 18 years were eligible. Before accessing the questionnaire, participants reviewed an on-screen information sheet describing study aims, procedures, data handling and confidentiality. Written electronic consent was obtained by ticking a mandatory check-box; continuation was impossible without consent and the timestamp was stored for auditing. No personal identifiers were collected, and respondents could withdraw at any time by closing the browser window. The questionnaire was distributed via a sponsored social media campaign—on Instagram and Facebook—managed by an agency that optimized the promotion of the survey through targeted advertisements. The questionnaire was structured into multiple dimensions. The first dimension gathered socio-demographic information; the second focused on dietary habits, body weight, and health-related aspects. The subsequent dimensions explored: attitudes toward organic foods, food origin and nutritional labeling; frequency of consumption of specific foods and perceived health impact of nutrients; cooking frequency, methods of food preparation, use of condiments, and wine consumption; food purchasing and consumption habits; adherence to both the nutritional and cultural components of the Mediterranean diet; and the relationship between food and sociality. To measure food-related behaviors, batteries of frequency scales were used, while Cantril and Likert scales were adopted to assess attitudes toward specific foods.

The questionnaire was pre-tested on a small sample of 20 subjects to identify any ambiguities or issues in question formulation; feedback received allowed for refinement of the final structure. The average completion time was approximately 15 minutes.

The survey advertisement generated a total of 267,775 impressions, reaching 86,636 unique users and obtaining 2,325 clicks (CTR: 0.87%). The most responsive segment consisted of individuals over the age of 50 residing in Central Italy (CTR: 1.30%). In total, 1,114 users accessed the questionnaire page; of these, 29 did not provide informed consent, while 966 completed the questionnaire.

The choice to employ a social network-based recruitment strategy, while cost-effective, presents several limitations, such as self-selection bias, potential exclusion of less digitally connected population groups (digital divide), and limited representativeness. Although a quota sampling logic was adopted to ensure a balance in terms of gender and geographical area, the final sample was strongly skewed by gender (Female: 88.3%; Male: 11.7%).

To support the analyses, several composite indices were constructed using additive or typological procedures, by recoding and aggregating the following variables: family cultural capital (educational attainment of mother and father); weekly consumption of processed foods (frequency of consumption of pre-cooked/frozen foods, fast food, sweets, sugary drinks, etc.); use of unhealthy cooking and seasoning techniques (frequency of frying, sautéing, use of butter/lard, etc.); use of raw seasoning (frequency of raw olive oil use); adherence to the Mediterranean diet both in its nutritional and cultural components (frequency and type of consumption of cereals, fruits, vegetables, legumes, olive oil, fish, meat, dairy products, sweets, along with questions on traditions and values); and Body Mass Index (BMI), based on self-reported weight and height. These indices enabled a more detailed description and comparison of dietary behaviors within the sample.

Data analysis was performed using SPSS statistical software. A progressive analytical approach was adopted. Univariate analyses were used to describe the composition of the sample and the main variables of interest (socio-demographic variables, dietary habits, lifestyles, etc.). Subsequently, bivariate analyses, particularly ANOVA, were conducted to explore differences in mean scores of Mediterranean diet adherence and other constructed indices across various socio-demographic and socio-economic characteristics (gender, age, educational attainment, geographical area, family cultural capital, BMI, etc.).

Consistent with the approach of integrated analytical models [43], multiple linear regression was employed to capture the interaction between structural and subjective dimensions in adherence to the Mediterranean diet. This technique made it possible to highlight how social inequalities are reflected in dietary behaviors, framing the Mediterranean diet not merely as a nutritional model but as a complex indicator of individual health and social position.

## 3. Sample characteristics and indices

The sample collected through the aforementioned recruitment strategy is predominantly female (88.3%) and mainly falls within the 18–35 age group (44.3%), followed by respondents aged 36–55 years (39.6%); approximately 16% of the sample is over 55 years of age. The majority of participants possess a medium to high level of education. More specifically, only 0.6% hold a primary school certificate and 16% have completed secondary education (high school diploma). The remainder of the sample holds either a three-year university degree (approximately 36%), a master's degree (40.4%), or a post-graduate qualification (approximately 24%). Furthermore, the sample is fairly evenly distributed across geographical areas: 28% reside in the North-West, followed by 29% in the South and Islands, 21.5% in the Centre, and 19.9% in the North-East. With regard to household monthly income, approximately 60% of the sample reported a net household income of around €3,000 per month. A summary is provided below (Table 1):

To provide a concise reconstruction of the occupational profile of workers (both employees and the self-employed), a typological index was developed [44] by combining two recategorized variables into distinct levels. Approximately 46% of workers hold positions related to craftsmanship or clerical services, a figure that corresponds with the average level of household income observed in the sample. Following this, around 26% are entrepreneurs or managers, while about 5% are employed in manual or similar labor occupations (Table 2).

Family cultural capital [14] is also a typological index that reflects the cultural context in which the respondent is embedded, particularly through the synthesis of the parents' educational attainment. It emerges that approximately 39% of the

**Table 1. Socio-demographic characteristics of the sample.**

| *Gender* | Female (88.3%); Male (11.7%) |
|---|---|
| *Age* | 18-35 years (44.5%); 36–55 years (39.6%); Overs 55 years (15.6%); Missing (0.2%) |
| *Educational Attainment* | Lower secondary school certificate (0.6%); Upper secondary or vocational school diploma (15.7%); Bachelor's degree (3-year university degree) (19.5%); Master's degree (40.4%); Post – Graduate degree (23.8%) |
| *Geographical Area* | North-West (28.5%); North-East (19.9%); Centre (21.5%); South and Island (29%). Missing (1.1%) |
| *Monthly Household Income* | Ups 1,500 euro (9.6%); 1,501–2,500 (20.1%); 2.501–3,000 (12.3%); 3,001–4,000 (14.5%); 4.001–5.000 (6.9%); Over 5,000 (8.1%); Missing (28.5%) |

*Source: Authors' elaboration based on personal data.*

**Table 2. Workers' occupational profiles.**

| Workers' Occupational Profiles | Frequency | Percentage |
|---|---|---|
| Executives or entrepreneurs | 249 | 25.8 |
| Craftsmen or clerical workers | 441 | 45.7 |
| Manual workers or collaborators | 48 | 5 |
| Total | 738 | 76.4 |
| Missing | 228 | 23.6 |
| Total | 966 | 100 |

*Source: Authors' elaboration based on personal data.*

sample (Table 3) has a medium level of family cultural capital; following this, around 24% of the sample is characterized by low cultural capital; finally, 22.3% is characterized by high cultural capital.

After reconstructing the cultural context of belonging, an objective health indicator was integrated, namely the Body Mass Index (BMI), calculated based on self-reported weight and height data. The index was calculated using the following formula:

$$BMI = \left( Weight \div Height^2 \right) [45]$$

Overall, from an initial categorization, approximately 69% of the sample falls within the normal weight range; about 6% are underweight; 3.5% are classified as having class I obesity; 1.3% have class II obesity; and 0.2% are in the highest obesity category (Table 4).

Respondents were then asked to rate on a scale from 1 to 10 the influence that diet and body weight have on a person's health status (see Tables 5 and 6). Approximately 73% of the sample attributed high importance to the role of diet in overall health.

On the other hand, respondents tend to attribute less importance to the role of body weight in the same relationship. As shown in Table 6, only 58% of the sample consider weight to have a significant impact on health status.

The final index constructed concerns adherence to the Mediterranean diet: a dietary pattern widely recognized for its beneficial effects on health [18,19,46,47]. This index was developed by the University of Rome "Foro Italico", in

**Table 3. Family cultural capital.**

| Family Cultural Capital | Frequency | Percentage | Cumulative Valid Percentage |
|---|---|---|---|
| Low family cultural capital | 231 | 23.9 | 28.1 |
| Medium family cultural capital | 376 | 38.9 | 73.8 |
| High family cultural capital | 215 | 22.3 | 100 |
| Total | 822 | 85.1 | |
| Missing | 144 | 14.9 | |
| Total | 966 | 100 | |

*Source: Our elaboration on personal data.*

**Table 4. Respondents by BMI.**

| Respondents by BMI | BMI range (kg/m2) | Frequency | Valid Percentage | Cumulative Percentage |
|---|---|---|---|---|
| Severe underweight | < 16.5 | 3 | 0.5 | 0.5 |
| Underweight | da 16.5 a 18.4 | 34 | 5.7 | 6.2 |
| Normal weight | da 18.5 a 24.9 | 413 | 68.8 | 75 |
| Overweight | da 25 a 30 | 120 | 20 | 95 |
| Class I obesity | da 30.1 a 34.9 | 21 | 3.5 | 98.5 |
| Class II obesity | da 35 a 40 | 8 | 1.3 | 99.8 |
| Class III obesity | > 40 | 1 | 0.2 | 100 |
| Total | | 600 | 100 | |
| Missing | | 366 | | |
| Total | | 966 | | |

*Source: Our elaboration based on personal data.*

**Table 5. Importance attributed to diet on health status, by levels.**

**In your opinion, on a scale from 1 to 10, where 1 means "not at all" and 10 means "very much," how much does diet influence a person's health?**

| | | Frequency | Percentage | Valid Percentage | Cumulative Percentage |
|---|---|---|---|---|---|
| Valid | Medium/High Weight | 203 | 21 | 27.4 | 27.4 |
| | High Weight | 539 | 55.8 | 72.6 | 100 |
| | Total | 742 | 76.8 | 100 | |
| Missing | System | 224 | 23.2 | | |
| Total | | 966 | 100 | | |

*Source: Our elaboration on personal data.*

collaboration with the University of Piemonte Orientale, and is inspired by validated tools in the literature, such as the score proposed by Trichopoulou [48] and the more recent Medi-Lite score elaborated by Sofi [49], both extensively used in epidemiological and clinical settings. Using an additive procedure, a score of 1 was assigned to the response identified by the medical unit as most consistent with the Mediterranean diet, and a score of 0 to all other responses, specifically:

Adherence is considered high when respondents report daily consumption of cereals and potatoes during main meals (approximately 80 g of cereals and/or 50 g of bread), along with adequate quantities of fruits and vegetables (about 150 g and 200 g, respectively). Frequent consumption of nuts, a daily intake of about 40 g of olive oil — the primary fat source

**Table 6. Importance attributed to body weight on health status by levels.**

In your opinion, on a scale from 1 to 10, where 1 means "not at all" and 10 means "very much," how much does body weight influence health?

|  |  | Frequency | Percentage | Valid Percentage | Cumulative Percentage |
|---|---|---|---|---|---|
| Valid | Medium/High Weight | 312 | 32.3 | 42 | 42 |
|  | High Weight | 430 | 44.5 | 58 | 100 |
|  | Total | 742 | 76.8 | 100 |  |
| Missing | System | 224 | 23.2 |  |  |
| Total |  | 966 | 100 |  |  |

*Source: Our elaboration on personal data.*

in the Mediterranean diet — and regular weekly consumption of milk or yogurt (3–4 times, 125 ml), white meats (100 g), fish or shellfish (150 g) are also reported. Foods such as cheeses, red meats, eggs, cured meats, and sweets should be consumed in moderation, ideally 1–2 times per week. Lastly, proper hydration is considered an integral part of the model: adherence includes consuming at least eight glasses of water per day.

Once constructed, the index was divided into adherence levels (Table 7). Results show that over 70% of the sample exhibits medium or low adherence to the Mediterranean diet, while only about 27% consistently follow this dietary pattern.

## 4. Determinants of adherence to the mediterranean diet

Given the nature of the collected data and the construction of the various indices presented in the previous section, multiple linear regression analysis was adopted as the appropriate statistical model to explore the relationship between the level of adherence to the Mediterranean diet (dependent variable) and all identified possible factors (independent variables) [50]. The relationship is expressed as follows:

$$Y = f(X_1, X_2, ..., X_m) + \varepsilon = f(X) + \varepsilon$$

Specifically, four incremental models were constructed (Table 8), based on dimensions identified in the literature: socio-demographic, economic and cultural factors; family composition; health status (BMI); presence of family members with obesity; awareness of different food nutrients; and perceptual variables concerning the influence of weight and diet on health.

The constructed models are sequential and integrated [18,43–51], allowing an understanding of how the weight and significance of factors change as new variables are introduced. A particularly relevant element within the models presented is the integration of variables typically of medical nature with those of a social nature. Following the literature on social determinants of health [1,2] the dependent variable in these models (adherence to the Mediterranean diet) is medical in nature, whereas the independent variables are social, with the exception of BMI. This modeling approach clarifies how social characteristics influence the adoption and adherence to the Mediterranean diet—a well-established parameter for assessing individual health status—and contributes to understanding the factors that promote or hinder a "healthy" lifestyle.

In the first model (Table 9), all variables related to the socio-demographic dimension were included, specifically gender (male/female), age (18–35 years, 36–55 years, over 55 years), educational level (up to high school diploma, bachelor's degree, master's degree, post-graduate degree), and geographical area of residence (North, Central, and Southern Italy). In the model presented here, as well as in the following models, the statistically significant regressors are reported in bold, with the reference categories indicated in parentheses. All models are weighted for gender and age. Regarding this dimension, the results indicate that all other factors being equal, the variables that influence adherence to the Mediterranean diet, in order of weight (Beta coefficient), are geographical area, age, and educational level. More specifically, higher

**Table 7. Levels of adherence to the mediterranean diet.**

**Levels of Adherence Index to the Mediterranean Diet**

| | Frequency | Percentage | Valid Percentage | Cumulative Percentage |
|---|---|---|---|---|
| Low adherence | 152 | 15.7 | 22.3 | 22.3 |
| Medium adherence | 343 | 35.5 | 50.4 | 72.7 |
| High adherence | 186 | 19.3 | 27.3 | 100 |
| Total | 681 | 70.5 | 100 | |
| Missing | 285 | 29.5 | | |
| Total | 966 | 100 | | |

*Source: Our elaboration based on personal data.*

**Table 8. Descriptive statistics of the four regression models.**

| Model | R | R-squared | Adjusted R-squared | Standard error of the estimate | Durbin-Watson |
|---|---|---|---|---|---|
| 1 | .186a | 0.035 | 0.023 | 0.196 | 1.982 |
| 2 | .225b | 0.051 | 0.026 | 0.195 | 1.970 |
| 3 | .249c | 0.062 | 0.028 | 0.195 | 1.980 |
| **4** | **.311d** | **0.263** | **0.251** | **1.193** | **1.989** |

*Source: Our elaboration based on personal data.*

**Table 9. Model 1.**

| Model 1 | Unstandardized Coefficients | | Standardized Coefficients | T | Sign. |
|---|---|---|---|---|---|
| | B | Standard Error | Beta | | |
| (Constant) | 0.503 | 0.030 | | 16.789 | 0.000 |
| Female (Male) | 0.019 | 0.023 | 0.031 | 0.815 | 0.415 |
| **Aged 18–35** | **0.041** | **0.017** | **0.142** | **2.437** | **0.015** |
| Aged 36–55 (Over 55 years) | 0.026 | 0.017 | 0.090 | 1.560 | 0.119 |
| Bachelor's Degree | 0.011 | 0.019 | 0.032 | 0.599 | 0.550 |
| Master's Degree | 0.019 | 0.017 | 0.066 | 1.166 | 0.244 |
| **Postgraduate Degree** (Up to High School Diploma) | **0.043** | **0.018** | **0.127** | **2.358** | **0.019** |
| **Northern Italy** | **−0.041** | **0.014** | **−0.144** | **−2.973** | **0.003** |
| Southern Italy (Central Italy) | −0.023 | 0.016 | −0.071 | −1.452 | 0.147 |

*Source: Our elaboration based on personal data; Dependent variable of the model: adherence to the Mediterranean diet.*

adherence is observed among younger individuals aged between 18 and 35 years, as well as among those with a higher educational qualification (a direct positive relationship between adherence and respondents holding a post-graduate degree). Conversely, those living in Northern Italy show lower adherence compared to those residing in Central and Southern Italy (an inversely proportional relationship).

In the second model (Table 10), in addition to the basic socio-demographic variables (gender, age, educational attainment, and geographical area of residence), additional socio-economic dimensions were introduced. Specifically:

**Table 10. Model 2.**

| Model 2 | Unstandardized Coefficients | | Standardized Coefficients | t | Sign. |
|---|---|---|---|---|---|
| | B | Standard Error | Beta | | B |
| (Constant) | 0.514 | 0.035 | | 14.865 | 0 |
| Female (Male) | 0.021 | 0.023 | 0.035 | 0.911 | 0.363 |
| **Aged 18–35** | **0.035** | **0.018** | **0.119** | **1.926** | **0.055** |
| Aged 36–55 (Over 55 years) | 0.024 | 0.017 | 0.081 | 1.367 | 0.172 |
| Bachelor's Degree | 0.003 | 0.019 | 0.008 | 0.157 | 0.875 |
| Master's Degree | 0.009 | 0.017 | 0.03 | 0.506 | 0.613 |
| **Postgraduate Degree** (Up to High School Diploma) | 0.030 | 0.02 | 0.087 | 1.502 | 0.134 |
| **Northern Italy** | **−0.048** | **0.014** | **−0.166** | **−3.399** | **0.001** |
| Southern Italy (Central Italy) | −0.021 | 0.016 | −0.064 | −1.285 | 0.199 |
| Occupational position related to entrepreneurship and managerial profiles | −0.004 | 0.016 | −0.012 | −0.251 | 0.802 |
| Occupational position related to craftsmanship or clerical services (Occupational position related to manual labor or similar activities) | −0.009 | 0.014 | −0.03 | −0.623 | 0.533 |
| Up to 1,500 euro | 0.002 | 0.02 | 0.003 | 0.08 | 0.936 |
| Between 1,501–2,500 | 0.024 | 0.016 | 0.069 | 1.506 | 0.133 |
| Between 2,501–3,000 | 0.008 | 0.019 | 0.019 | 0.432 | 0.666 |
| **Between 3,001–4,000** | **0.038** | **0.017** | **0.101** | **2.214** | **0.027** |
| **Between 4,001–5,000** (Over 5,000 euro) | **0.048** | **0.022** | **0.093** | **2.189** | **0.029** |
| Low family cultural capital | −0.023 | 0.016 | −0.074 | −1.464 | 0.144 |
| Medium family cultural capital (High family cultural capital) | −0.006 | 0.014 | −0.021 | −0.451 | 0.652 |

*Source: Our elaboration on personal data; Dependent variable of the model: adherence to the Mediterranean diet.*

occupational position (distinguishing between positions related to entrepreneurship and management profiles; positions related to craftsmanship or clerical services; positions related to manual labor or similar activities), net monthly household income (categorized as between €1,501 and €2,500; €2,501 and €3,000; €3,001 and €4,000; €4,001 and €5,000; and over €5,000), and family cultural capital (low, medium, or high family cultural capital), with the aim of examining potential differences.

In this model, it emerges that, all other factors being equal, adherence to the Mediterranean diet is influenced by age, geographical area, and net monthly household income. The most substantial influence is attributed to geographical area: as in the previous model, Northern Italy is shown to have a lower adherence to the Mediterranean diet compared to other regions of the country. Secondly, age appears to be a relevant factor: individuals aged between 18 and 35 years demonstrate greater adherence compared to the other age groups. Finally, household income has an effect: specifically, a net monthly income above €3,000 is positively associated with higher adherence to the Mediterranean diet.

This latter variable seems to "replace" the role previously played by high educational attainment, giving greater weight to economic resources rather than to institutionalized cultural capital [52,53].

In the third model (Table 11), in addition to the variables related to the socio-demographic dimension, occupational position, income, and basic family cultural capital, new dimensions concerning the family context and the respondent's physical condition were introduced. Specifically, variables related to household composition in terms of the presence and/or number of children (no children, one child, two children, more than two children), the age group of the children (recoded

**Table 11. Model 3.**

| Model 3 | Unstandardized Coefficients | | Standardized Coefficients | t | Sign. |
|---|---|---|---|---|---|
| | B | Standard Error | Beta | | |
| (Constant) | 0.522 | 0.050 | | 10.495 | 0 |
| Female (Male) | 0.017 | 0.023 | 0.027 | 0.707 | 0.480 |
| **Aged 18–35** | 0.027 | 0.026 | 0.092 | 1.031 | 0.303 |
| Aged 36–55 (Over 55 years) | 0.015 | 0.025 | 0.051 | 0.603 | 0.547 |
| Bachelor's Degree | 0.000 | 0.020 | 0.000 | −0.005 | 0.996 |
| Master's Degree | 0.007 | 0.017 | 0.023 | 0.389 | 0.698 |
| **Postgraduate Degree** (Up to High School Diploma) | 0.028 | 0.020 | 0.083 | 1.426 | 0.154 |
| **Northern Italy** | **−0.049** | **0.014** | **−0.169** | **−3.426** | **0.001** |
| Southern Italy (Central Italy) | −0.020 | 0.016 | −0.062 | −1.239 | 0.216 |
| Occupational position related to entrepreneurship and managerial profiles | −0.007 | 0.016 | −0.020 | −0.403 | 0.687 |
| Occupational position related to craftsmanship or clerical services (Occupational position related to manual labor or similar activities) | −0.007 | 0.014 | −0.024 | −0.503 | 0.615 |
| Up to 1,500 euro | 0.002 | 0.020 | 0.005 | 0.109 | 0.913 |
| Between 1,501–2,500 | 0.025 | 0.016 | 0.073 | 1.584 | 0.114 |
| Between 2,501–3,000 | 0.005 | 0.019 | 0.013 | 0.290 | 0.772 |
| **Between 3,001–4,000** | **0.039** | **0.017** | **0.104** | **2.253** | **0.025** |
| **Between 4,001–5,000** (Over 5,000 euro) | **0.047** | **0.022** | **0.090** | **2.081** | **0.038** |
| Low family cultural capital | **−0.026** | **0.016** | **−0.082** | **−1.613** | **0.007** |
| Medium family cultural capital (High family cultural capital) | −0.006 | 0.014 | −0.021 | −0.443 | 0.658 |
| No Children | 0.012 | 0.040 | 0.041 | 0.312 | 0.755 |
| Only One Child | 0.028 | 0.034 | 0.076 | 0.834 | 0.405 |
| Two Children (More than two Children) | 0.016 | 0.034 | 0.041 | 0.482 | 0.603 |
| With minor Children (With adult Children) | 0.013 | 0.028 | 0.038 | 0.461 | 0.645 |
| People without obesity in the family (People with obesity in the family) | −0.021 | 0.018 | −0.045 | −1.169 | 0.243 |
| Normal weight | 0.016 | 0.013 | 0.055 | 1.212 | 0.226 |
| Overweight (Underweight) | −0.012 | 0.017 | −0.033 | −0.732 | 0.465 |

*Source: Our elaboration based on personal data; Dependent variable: adherence to the Mediterranean diet.*

as adult children and minor children), the presence of obese family members, and the respondent's specific physical condition, recoded based on BMI (underweight, normal weight, overweight), were included.

By incorporating these new dimensions into the study, a particularly interesting shift emerges. Holding other factors constant, respondents with higher income, residing outside Northern Italy, and possessing high cultural capital show greater adherence to the Mediterranean diet. Notably, young age, which was significant in the previous model, no longer remains so. Instead, when considering specific family context variables, cultural capital assumes a central role.

Thus, this model further confirms the pivotal influence of cultural dimension in shaping dietary behaviors. The symbolic and relational environment in which individuals grow up appears to exert a deeper influence than mere socio-demographic conditions or individual health status. This reinforces the hypothesis that adherence to healthy eating practices is strongly mediated by embodied cultural capital transmitted at the family level [52].

The fourth and final model (Tables 12 and 13) is configured as the "optimal model" ($R^2 = 25\%$; Durbin-Watson = 1.98) because it integrates all dimensions and explains a substantial proportion of the variance. Parameters used: Tolerance is defined as $1 - R^2\_i0$, and the Variance Inflation Factor (VIF) is given by $VIF\_i = 1/(1 - R^2\_i0)$, where $R^2\_i0$ represents the squared coefficient measuring the correlation between the j-th explanatory variable in the model and all other explanatory variables. A VIF greater than 5 and an $R^2\_i0$ value close to 1 are indicators of high multicollinearity. Therefore, after several iterations and multicollinearity analyses, the tested set of variables is as follows:

Compared to the other models, the new dimensions added (Tables 12 and 13) concern the level of awareness regarding the consumption of various macronutrients (sugars, fats, carbohydrates, fructose, proteins, and fiber), the perception of the role of nutrition on people's health, and the perception of the role of body weight on health. The model remains stable compared to the previous one; however, in addition to income, the type of perception regarding the influence of nutrition and weight on health plays a central, significant, and directly proportional role (with Northern Italy residence and low cultural capital still inversely related to adherence to the Mediterranean diet). In fact, in terms of Beta coefficients, these two factors are among those that most strongly influence adherence to the Mediterranean diet, confirming the cultural aspect of the dietary choices adopted. Thus, adherence to the Mediterranean diet appears to be connected not only to

**Table 12. Analytical description of model 4.**

| Y:Adherence to the Mediterranean diet |
|---|
| X1: Gender |
| X2: Age |
| X3: Educational attainment |
| X4: Geographic area of residence |
| X5: Occupational position |
| X6: Monthly family income |
| X7: Family cultural capital |
| X8: Number of children |
| X9: Age group of children |
| X10: Presence of individuals with obesity in the family |
| X11: BMI |
| X12: Level of awareness regarding the consumption of main macronutrients present in foods (sugars, fats, proteins, fructose, carbohydrates, and fiber) |
| X13: Perception of the role of nutrition on individual health |
| X14: Perception of the role of body weight on individual health |
| e: residual/error term |

*Source: Our elaboration based on personal data.*

**Table 13. Model 4.**

| Model 4 | Unstandardized Coefficients | | Standardized Coefficients | t | Sign. |
|---|---|---|---|---|---|
| | B | Standard Error | Beta | | |
| (Constant) | 0.564 | 0.070 | | 8.077 | 0 |
| Female<br>(Male) | 0.010 | 0.023 | 0.017 | 0.436 | 0.663 |
| **Aged 18–35** | 0.033 | 0.026 | 0.113 | 1.267 | 0.206 |
| Aged 36–55<br>(Over 55 years) | 0.019 | 0.025 | 0.065 | 0.77 | 0.442 |
| Bachelor's Degree | −0.001 | 0.020 | −0.003 | −0.061 | 0.952 |
| Master's Degree | 0.004 | 0.017 | 0.015 | 0.258 | 0.797 |
| **Postgraduate Degree**<br>(Up to High School Diploma) | 0.028 | 0.020 | 0.083 | 1.432 | 0.153 |
| **Northern Italy** | **−0.048** | **0.014** | **−0.167** | **−3.393** | **0.001** |
| Southern Italy<br>(Central Italy) | −0.020 | 0.016 | −0.061 | −1.217 | 0.224 |
| Occupational position related to entrepreneurship and managerial profiles | −0.009 | 0.016 | −0.029 | −0.578 | 0.563 |
| Occupational position related to craftsmanship or clerical services (Occupational position related to manual labor or similar activities) | −0.008 | 0.014 | −0.027 | −0.566 | 0.572 |
| Up to 1,500 euro | 0.005 | 0.020 | 0.012 | 0.274 | 0.784 |
| Between 1,501–2,500 | 0.028 | 0.016 | 0.081 | 1.768 | 0.078 |
| Between 2,501–3,000 | 0.014 | 0.019 | 0.033 | 0.741 | 0.459 |
| **Between 3,001–4,000** | **0.045** | **0.017** | **0.118** | **2.575** | **0.010** |
| **Between 4,001–5,000**<br>(Over 5,000 euro) | **0.048** | **0.022** | **0.092** | **2.144** | **0.032** |
| Low family cultural capital | **−0.029** | **0.016** | **−0.091** | **−1.792** | **0.024** |
| Medium family cultural capital<br>(High family cultural capital) | −0.008 | 0.014 | −0.026 | −0.543 | 0.587 |
| No Children | 0.004 | 0.040 | 0.012 | 0.092 | 0.927 |
| Only One Child | 0.015 | 0.034 | 0.040 | 0.436 | 0.663 |
| Two Children<br>(More than two Children) | 0.006 | 0.034 | 0.014 | 0.162 | 0.871 |
| With minor Children<br>(With adult Children) | 0.013 | 0.028 | 0.039 | 0.476 | 0.634 |
| People without obesity in the family<br>(People with obesity in the family) | −0.022 | 0.018 | −0.048 | −1.223 | 0.222 |
| Normal weight | 0.013 | 0.013 | 0.044 | 0.962 | 0.336 |
| Overweight<br>(Underweight) | −0.011 | 0.017 | −0.029 | −0.640 | 0.523 |
| Awareness level regarding the consumption of sugar | −0.003 | 0.004 | −0.033 | −0.730 | 0.466 |
| Awareness level regarding the consumption of fats | 0.006 | 0.003 | 0.086 | 1.898 | 0.058 |
| Awareness level regarding the consumption of carbohydrates | −0.001 | 0.003 | −0.021 | −0.419 | 0.676 |
| Awareness level regarding the consumption of fructose | 0.003 | 0.003 | 0.043 | 0.893 | 0.372 |
| Awareness level regarding the consumption of proteins | 0.000 | 0.003 | 0.002 | 0.041 | 0.967 |

*(Continued)*

**Table 13.** (Continued)

| Model 4 | Unstandardized Coefficients | | Standardized Coefficients | t | Sign. |
|---|---|---|---|---|---|
| | B | Standard Error | Beta | | |
| Awareness level regarding the consumption of fibers | −0.003 | 0.003 | −0.057 | −1.09 | 0.276 |
| **Perception of the role of nutrition on a person's health** | **−0.018** | **0.007** | **0.133** | **2.819** | **0.005** |
| **Perception of the role of body weight on a person's health** | **0.012** | **0.005** | **0.109** | **2.326** | **0.020** |

*Source: Our elaboration based on personal data; Dependent variable of the model: adherence to the Mediterranean diet.*

structural material and cultural resources but also to the subjective capacity to assign meaning and value to food choices, consistent with a multidimensional view of the determinants of health.

## 5. Study limitations

This research utilizes non-probability sampling through sponsored advertisements on social media, which may lead to selection bias and issues related to the digital divide, such as the exclusion of groups with limited digital access. Despite the intention to use quota logic to ensure a balance in gender and geographic representation, the recruitment process resulted in a sample predominantly composed of females (88.3%). We intentionally opted not to adjust quotas after the fact (such as through post-stratification or weighting) to gauge natural interest in the subject among different genders. However, this choice restricts representativeness and suggests that gender comparisons should be viewed as descriptive rather than as inferences applicable to the entire population.Additionally, age diversity was limited, further limiting the generalizability to the broader Italian population. Campaign analytics indicated differential engagement (e.g., higher CTR among adults aged 50+ in Central Italy), which may have shaped the respondent profile.It is crucial to recognize that the web-based element discussed here is just one aspect of a larger, ongoing initiative that also involves offline recruitment in clinical environments. The data gathered from a particular patient group in partnership with collaborating laboratories are not part of this manuscript. Consequently, the current findings should be viewed as exploratory and supplementary within the context of a more extensive research program.

Regarding the validity of the data, since the information was self-reported, it may be affected by recall bias, as participants' responses rely on their ability to accurately remember and describe their dietary habits and cooking methods. In similar research, food diaries are occasionally used to minimize recall mistakes; however, in our situation, this tool was not suitable for the web-based protocol we employed. We sought to mitigate recall risk using clear, specific items anchored to recent behaviors, standardized response scales, and anonymous online administration. For some variables, item non-response exceeded 10%. To avoid introducing non-verifiable assumptions about the missing-data mechanism, we did not apply imputation; analyses used pairwise deletion and reported the item-level valid N. Overall, these limitations recommend caution in generalizing beyond the study sample while still providing proper, internally coherent evidence to inform subsequent phases of the DIET-FRUCTOSE project.

## 6. Discussion and Conclusion

Adherence to the Mediterranean diet, understood as a healthy and culturally rooted dietary practice, has been analyzed through an incremental approach based on four linear multiple regression models [18,43–51]. This methodological choice responds to the need to progressively and structurally explore the influence exerted by variables of different nature—socio-demographic, economic, cultural, familial, health-related, and perceptual—on the dependent variable, namely the level of adherence to the Mediterranean diet.

The analytical design adopted aligns with a broad theoretical framework (illustrated in the first part), which recognizes eating practices as the result of a multi-level system of determinants—individual, relational, and structural. In particular, the model of social determinants of health proposed by Dahlgren and Whitehead [1] provides the interpretative framework for analyzing inequalities in eating behaviors, highlighting how socioeconomic, cultural, and environmental variables systematically influence health. At the same time, Popkin's [6] theory of nutritional transition clarifies how changes in contemporary dietary regimes are linked to macro-social transformations, including urbanization, food industrialization, and globalization of consumption. Finally, Bourdieu's sociological approach [5] allows us to interpret food choices not only as functional practices but also as symbolic acts, stratified along lines of economic and cultural capital.

The approach adopted enables not only the evaluation of the direct effect of each factor but also the observation of how these effects change with the introduction of new explanatory dimensions, in line with the model of social determinants of health [1,2].

In the first model, the analysis focuses on demographic and territorial variables, such as gender, age, educational attainment, and geographic area of residence. Results show that young adults (18–35 years) and subjects with a postgraduate degree exhibit greater adherence to the Mediterranean diet. Conversely, living in Northern Italy is associated with lower adherence levels. These findings suggest that age and education—understood as indicators of cultural openness and predisposition toward healthy lifestyles—play a relevant role, while territorial differences may reflect specific cultural and dietary habits of different regions.

With the introduction in the second model of variables related to occupational status, family income, and family cultural capital, the analysis enriches with a more articulated socio-economic dimension. In this context, a monthly family income above 3,000 euros emerges as a significant positive predictor of adherence to the Mediterranean diet, while the effect of educational attainment tends to lose significance. With the inclusion of these new variables, economic capital appears to take precedence over institutionalized cultural capital, suggesting that the concrete possibility of accessing healthy and quality foods is strongly conditioned by available economic resources [52]. Residence in Northern Italy continues to be negatively associated with adherence, confirming a stable territorial pattern.

The third model introduces variables related to family composition (number and age group of children), the presence of obese family members, and the respondent's weight status (BMI). However, these dimensions do not show significant association with adherence to the Mediterranean diet. Conversely, the role of income strengthens and the importance of family cultural capital reemerges: lower levels of the latter are associated with lower adherence. This suggests that, beyond family conditions or individual physical status, the cultural environment in which one was raised exerts a profound influence on food choices. These results confirm that eating is not only a physiological practice but also a stratified cultural and social fact, as maintained by Bourdieu [52]. Family cultural capital acts as a symbolic mediator in access to quality dietary regimes, producing effects that go far beyond formal education or economic resources alone.

The fourth model represents the most complete configuration, also integrating perceptual and cognitive variables such as awareness of macronutrient consumption (sugars, fats, carbohydrates, fructose, fibers, and proteins) and perception of the role of nutrition and body weight on health. This model, which explains 25% of the variance ($R^2 = 0.25$) and shows a good index of error independence (Durbin-Watson = 1.98), confirms the stability of previous results but introduces a crucial element: subjective perception. In particular, individuals who recognize the importance of nutrition and body weight for health show greater adherence to the Mediterranean diet. These perceptual variables rank among the most influential in terms of standardized coefficient (Beta), indicating that awareness and mental attitude toward health represent fundamental determinants.

Overall, the analysis highlights how adherence to the Mediterranean diet results from a complex interaction between economic, cultural, and perceptual factors. While demographic characteristics and educational attainment prevail in the initial models, income, family cultural capital, and finally individual awareness emerge more strongly in subsequent

models. This trajectory reflects an interpretative evolution moving from a structural to a more subjective and cultural reading of eating behavior.

It is not only economic availability that matters but also the awareness and symbolic value attributed to food, in a framework intertwining public health, culture, and inequalities.

In summary, adherence to the Mediterranean diet—as recently emphasized by the Italian National Institute of Health [46]—cannot be reduced solely to economic availability or formal education level, but is strongly influenced by cultural, cognitive, and perceptual factors. These findings confirm the need for integrated public policies that promote food education, equitable access to healthy foods, and the spread of a culture of health, with particular attention to geographic areas and vulnerable social groups.

Finally, eating—far from being a purely nutritional act—represents a concrete expression of the social, cultural, and symbolic inequalities permeating individuals' daily lives [46]. As Bourdieu [52] reminded us, even the simplest gesture—like choosing what to eat—embodies a social structure: in this sense, the Mediterranean diet is not just a health prescription but a social language reflecting belonging, resources, and worldviews. Understanding adherence to this dietary model thus means recognizing that health is built not only in bodies but also in material conditions, family ties, and cultural meanings that society attributes to food and "the good life."

## Supporting information

**S1 File.  This Excel file contains the dataset used for the analysis reported in the manuscript.** This Excel file contains the dataset used for the analysis reported in the manuscript. **Sheet 1 – Raw data:** anonymized responses collected in the survey. **Sheet 2 – Variables description:** list and definition of all variables included in the study.
(XLSX)

## Author contributions

**Conceptualization:** Francesca Romana Lenzi, Michela Cavagnuolo, Ciro Clemente De Falco, Ferdinando Iazzetta.

**Data curation:** Michela Cavagnuolo.

**Formal analysis:** Michela Cavagnuolo.

**Funding acquisition:** Francesca Romana Lenzi.

**Investigation:** Michela Cavagnuolo, Ferdinando Iazzetta.

**Methodology:** Francesca Romana Lenzi, Michela Cavagnuolo, Ciro Clemente De Falco, Vincenzo Esposito.

**Project administration:** Francesca Romana Lenzi.

**Resources:** Francesca Romana Lenzi.

**Supervision:** Francesca Romana Lenzi, Michela Cavagnuolo.

**Validation:** Francesca Romana Lenzi.

**Visualization:** Ciro Clemente De Falco, Vincenzo Esposito.

**Writing – original draft:** Francesca Romana Lenzi, Michela Cavagnuolo, Ciro Clemente De Falco, Ferdinando Iazzetta, Vincenzo Esposito.

**Writing – review & editing:** Francesca Romana Lenzi, Ciro Clemente De Falco, Vincenzo Esposito.

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
