## [Decision Letter · Decision Letter 0]

29 Jul 2025

Dear Dr. Lenzi,

Thank you for submitting your manuscript to PLOS ONE. After careful consideration, we feel that it has merit but does not fully meet PLOS ONE’s publication criteria as it currently stands. Therefore, we invite you to submit a revised version of the manuscript that addresses the points raised during the review process.

We look forward to receiving your revised manuscript.

Kind regards,

António Raposo

Academic Editor

PLOS ONE

 [The study was funded by Cariplo Foundation - Project Dietary fructose in pediatric obesity-related diseases: identification of nutritional, biological, omics and social determinants]. 

3. We note that your Data Availability Statement is currently as follows: [all relevant data are within the manuscript and its Supporting Information files.]

Additional Editor Comments (if provided):

Reviewers' comments:

Reviewer's Responses to Questions

**Comments to the Author**

1. Is the manuscript technically sound, and do the data support the conclusions?

Reviewer #1: Yes

Reviewer #2: Yes

2. Has the statistical analysis been performed appropriately and rigorously?

Reviewer #1: Yes

Reviewer #2: Yes

3. Have the authors made all data underlying the findings in their manuscript fully available?

Reviewer #1: Yes

Reviewer #2: Yes

4. Is the manuscript presented in an intelligible fashion and written in standard English?

Reviewer #1: Yes

Reviewer #2: Yes

Reviewer #1: 1. Please address the following biases in the manuscript which may affect the results:

a. The sample is gender skewed (88.3% females).

b. Selection bias due to reliance on social media recruitment predominantly.

c. The population selected is limited in age diversity, and therefore does not represent the big population.

d. Recall bias because of self reported data.

2. High rate of data missing with some variable such as BMI and income. How was the statistical analysis done with these missing data points.

3. Introduction should be trimmed. It is lengthy and theoretical.

4. Describe how adherence to Mediterranean diet was operationalized.

Reviewer #2: This is an interesting and well written manuscript addressing an important subject matter. The authors presented an in depth understandable and knowledge about the subject matter. The methodology was well written and explained and the results are novel.

**Do you want your identity to be public for this peer review?** For information about this choice, including consent withdrawal, please see our Privacy Policy

Reviewer #1: **Yes: ** Ankit Agarwal

Reviewer #2: No

---

## [Author Response · Author response to Decision Letter 1]

30 Sep 2025

Dear Editor and Reviewers,

Thank you for your thoughtful feedback—our revisions have benefited substantially from your comments.

We confirm full compliance with PLOS ONE style (incl. file naming) and have uploaded Response to Reviewers, Revised Manuscript with Track Changes, and Manuscript. The funders had no role in study design, data collection and analysis, decision to publish, or preparation of the manuscript. We have shared the complete minimal data set

Reviewer #1

1.Biases and validity concerns

We added a short integrative paragraph in the limitations section that directly addresses the methodological issues raised and clarifies our approach and rationale for key design choices. Specifically, the new paragraph is as follows.

• discusses the recruitment method based on social media that does not rely on probability, along with the consequences for selection bias and the digital divide;

• acknowledges the female-skewed sample and limited age diversity, clarifying that we intentionally did not apply post-hoc adjustments to observe natural interest by gender and that comparisons are descriptive;

• notes the self-reported nature of measures and how we mitigated recall risk.

• frames the web-based component as one part of a broader, ongoing program, positioning the current results as exploratory.

2. Handling of missing data

In the same integrative paragraph, we clarified that we used pairwise deletion, and reported item-level valid N for transparency. This explains our analytic choices and their rationale without overextending claims.

3. Introduction is lengthy

We streamlined the Introduction, focusing on the rationale, objectives, and key hypotheses and removing non-essential theoretical digressions.

4. Operationalization of Mediterranean-diet adherence

We have now stated concisely how adherence is defined and scored, so that readers can grasp the index without a long methodological detour.

Reviewer #2

We appreciate this positive appraisal. We have made minor editorial refinements and incorporated the clarifications above to enhance transparency and interpretability.

---

## [Editor Report · Decision Letter 1]

8 Oct 2025

Toward an Integrated Model of Dietary Behavior: Social Determinants and Adherence to the Mediterranean Diet

PONE-D-25-34789R1

Dear Dr. Lenzi,

We’re pleased to inform you that your manuscript has been judged scientifically suitable for publication and will be formally accepted for publication once it meets all outstanding technical requirements.

Kind regards,

António Raposo

Academic Editor

PLOS ONE
---

## [Editor Report · Acceptance letter]

PONE-D-25-34789R1

PLOS ONE

Dear Dr. Lenzi,

I'm pleased to inform you that your manuscript has been deemed suitable for publication in PLOS ONE. Congratulations! Your manuscript is now being handed over to our production team.

Kind regards,

on behalf of

Dr. António Raposo

Academic Editor

PLOS ONE